# Cultural Identity Performances on Social Media: A Study of Bolivian Students

**Paola Condemayta Soto \*** , **Joke Bauwens and Kevin Smets**

Department of Media and Communication Studies, Vrije Universiteit Brussel, 1050 Brussels, Belgium
* Correspondence: paola.norah.condemayta.soto@vub.be

**Abstract:** In this study, both performance and polymedia serve as important conceptual lenses to examine how university students in the Global South handle the social media landscape in enacting cultural identity. Based on 17 focus groups with 105 students from Bolivian universities, we argue that in performing their multiplex identities, this group of Bolivian young people navigate social media as polymedia environments, taking advantage of its possibilities and testing its constraints. The research generated three key findings: (1) students mainly reported examples of cosmopolitan and national identity performances; (2) performances of national belonging showed an ambiguous mixture of self-glorification and self-reflexivity; (3) indigenous identities were rarely performed on the platforms used.

**Keywords:** social media; cultural identity; performance studies; polymedia; Bolivia; university students

## 1. Introduction

> "I believe that social media influences you in a way, because through social media you can see the whole world or other cultures and traditions. Sometimes you can be influenced by that, so that you lose your identity because you want to be like (people) from other countries and see those things . . . ".

This statement from Carolina, a 23-year-old young woman from Bolivia, is a prime example of how university students in Global South countries experience the impact of popular social media communication on their non-Western cultural identities (Effiong 2018). With the massive impact of mobile communication on internet use in the Global South (Willems 2021), the question of how people in these parts of the world, without leaving their country, are expressing their cultural identity through social media communication enjoys a growing interest (Miller 2016; Sinanan 2020). Although the phrase "Global South" encompasses vast regions and numerous countries in Africa, Asia, Latin America and Oceania, general trends can be observed in this field of research. The shared history of colonialism, neo-imperialism, cultural and political marginalization, differential economic and social change, and related inequality (Dados and Connell 2012) provides a possible explanation for this. Generally, the question as to how in non-western cultures social media are reconfiguring the sense and concept of cultural identity is discussed with reference to (1) the ambiguous dynamics of the global–local nexus and, more specifically, the impact of global mass culture on traditional ways of living; (2) the communal meaning of cultural identity as opposed to the western individualistic concept of cultural selfhood; (3) concerns about lack of cultural values, knowledge and certainty, especially among the younger generations; (4) but also the potential of social media to rediscover, reinvigorate and popularize cultural traditions inside and outside of their countries' borders (e.g., Effiong 2018; Jakaza 2022; Uimonen 2013; Villacrés Roca 2016).

This article looks at some of these ideas in the specific context of Bolivia. With nearly 60% of its population younger than 29 years (INE 2017), a medium level of social media penetration among Latin American countries (Statista 2021) and its official status as a

pluricultural country, Bolivia offers a compelling context to study experiences of social media among university students outside the western world. Fully aware of the important line of research into Latin American people's social media use, and how this interacts with social protest, political participation and identity politics (e.g., Zegada and Guardia 2018), the present study assumes a more everyday, ethnographic approach to social media and cultural identity in Latin America. In doing so, it leans on anthropological and ethnographic scholarship "to understand ways of life as they are lived" (Hine 2017).

Starting from the idea that cultural identities are multiplex identities (Conquergood 1991), and viewing social media as polymedia environments (Madianou 2020), we examine how Bolivian university students perceive social media as yet another stage for representing their different cultural affinities and belongings. In line with our theoretical approach, we consider the context in which these social media are used to be critical (Madianou and Miller 2013; Miller 2016; Willems 2021). Within the contemporary Bolivian context, marked by external and internal colonialism, multiculturalism and involvement with nationhood (Albro 2010; Rasmussen 2021), we are particularly interested in the shifting forms of visibility at the nexus of globalization, consumer culture and social media.

Against this backdrop, this paper looks specifically at Bolivian young people studying at universities in Cochabamba, La Paz and Santa Cruz, and how they handle the social media landscape for enacting cultural identity. Based on 17 focus groups with 105 students, it explores how important it is for these students to show their cultural affinities and belongings on social media, and how social media are chosen as stages for expressing cultural identities. In the context of Bolivia, a country that has experienced a massive expansion of higher education and increasing access to it (Forste et al. 2004), students represent a relevant and relatively diverse demography in terms of regional, cultural, social and economic backgrounds (Peterson 2001). At the same time, both their educational and status attainment aspirations make this group of young people an interesting microcosm for studying the mechanisms that are at work in enacting and reproducing ideas about local culture, global citizenship, worldliness, cosmopolitanism, etc. (Lee 2022). After situating the particularities of the Bolivian context of the study, we describe the methodology of the focus group research. The subsections of the findings focus on general social media use, everydayness and normativity, cosmopolitanism, Bolivianness and indigenous identity.

## 2. Theoretical Framework

### 2.1. Social Media as Polymedia Performance Spaces

Since this approach starts from a user point of view, we understand social media as a wide range of online platforms that fulfill an important function in terms of identity, reputation, co-presence, connectivity, sociality and kinkeeping (Kietzmann et al. 2011; Ellison and Boyd 2013). Hence, microblogs, social networks, social gaming, instant messaging, photo-sharing and video-sharing are all seen as part of a variable and interrelated ensemble from which a person picks and combines to present to others, to manage relationships with others, to communicate with others. Research on social media and identity has demonstrated that young people invest a great deal of time and energy in identity work on social media by routinely and actively curating their identities via profiles (Thomas et al. 2017), images (Wargo 2017), language use (Lee 2014), geotagging (Schwartz and Halegoua 2015), nicknames (Shafie et al. 2012), privacy settings (Georgalou 2016; Hodkinson 2017), etc. In this regard, the notion of performance, as developed by Goffman (1959), has become pivotal to our understanding of how people represent aspects of their identity to others through social media. Just as offline social environments provide opportunities for people to present themselves to others, each social medium has its own platform specificities, mobilizing particular types of interaction (Papacharissi 2009) and appearances through different personas (van Dijck 2013b).

Thus far, as pointed out by Dumitrica (2019), the "relationship between social media mediation and cultural identity remains an under-studied problematic" (p. 304). Most work on cultural identity performance of young people on social media has been concerned with

migration, minorities, or activism (Mao and Shen 2015; Maragh 2018; Nziba Pindi 2018), but has failed to address cultural identity performances of young people who do not necessarily identify themselves as migrants or minorities. As we will see, their everyday uses of social media are imbued with complex performances of multifaceted cultural identities. Evidence on Latin American young people suggests that social media are viewed as spaces that "provide more freedom to embrace a range of identities" (Arends and Hordijk 2016, p. 18).

The idea of regarding cultural identity as a form of performance has become increasingly accepted in the literature (see Bauman 2000; Pfister and Hertel 2008; Whigham 2014). Leading authors from the fields of symbolic interactionism, anthropology, cultural studies and gender theory, respectively, such as Goffman (1959), Geertz (1973), Hall and du Gay (1996) and Butler (1990), have played a key role in this expansion. Rather than conceiving of cultural identity as an underlying essence or integral, inevitable nature that results in 'typical' cultural activities, it is understood as a "process of becoming rather than being" (Hall 1996, p. 4), performatively constituted by the very acts that are said to be its results. With the ambient presence of social media in young people's everyday life, understanding subjective identity in terms of performance has become even more focal in research (Cover 2012). Defined as inherently social, as it occurs "before a particular set of observers (...) which has some influence on the observers" (Goffman 1959, p. 13), the everydayness of presenting and performing oneself to others has culminated in an intensified navigation of online visibility, authenticity and self-promotion to known and unknown others.

Drawing on the fields of symbolic interactionism, anthropology, cultural studies and gender theory, performing also means that identity emerges in the doing and is therefore a process that is never completed. Although there is always the possibility of small deviations from established patterns that redefine the meaning of cultural identity, identity performance is often discussed in terms of recurrent practices (Butler 2011). A larger part of cultural identity performances actually consists of "shared, embodied practices that, through their enactment, represent the self-proclaimed cultural history of a group, oftentimes through the utilization of symbolic gestures or actions that are seen by the group as having a historical or traditional significance" (Whigham 2014, p. 204). Whigham argues that these cultural performances are important moments of collective identity construction, demonstrating to observers what the collective (i.e., group, community) believes to be important to share in a public, visible way. Subjective identity and identity performance are thus intertwined.

As culture can be construed as a fairly general term, referring to a multitude of elements (Williams 1976), individuals can draw from a vast range of resources. Cultural belonging is an equally general term and can be articulated through sports, music, food, language, traditional festivals, etc. For instance, Appadurai (1996) points out that certain sports, such as cricket, "come with a set of links between value, meaning, and embodied practice that are difficult to break and hard to transform" (p. 90). Leonard (2005) highlights singing and dancing as markers of Irish cultural identity, allowing people to perform their 'Irishness,' which operates as a public signal of identification. Other scholars have discussed the role of language (Bandhu 1989), food and cuisine (Imilan 2015) and urban folklore performances (Goldstein 1998). Recent studies show that on the Internet the same set of resources can be used in the process of performing cultural belonging. Studies on topics as diverse as language use (Han 2020), food-blogging (Solanilla and Medina 2016) and political activism (Ershov 2015; Halpern et al. 2017) show that social media in particular allow people to articulate cultural identity on a daily basis. The complex ways in which social media enable this require us to delve into the literature on social media as a space for identity performance.

Social media enable users to present themselves to others through their profiles, photos, nicknames, videos, status and stories. As a result, they provide and act as performance spaces where people can use online social networks such as sites of self-presentation and identity negotiation (Papacharissi 2012). In his dramaturgical account of human interaction, Goffman (1959) argued that we display a series of masks to others, enacting

roles, controlling and staging how we appear, ever concerned with how we are coming across, constantly trying to set ourselves in the best light. According to Goffman, people play different parts determined by situation. We are what we are, depending on the people we are interacting with. Goffman's dramaturgical understanding of everyday behavior has been applied to how people present and perform themselves online. A large number of studies specify how 'performances' given by individuals online often are crafted for particular audiences (Hogan 2010; Marwick and Boyd 2011), supporting the idea that users have agency over this networked space where the digital selfhood is merely relational (Campbell and Haynes 2020). It has therefore become commonplace to think of social media use as a type of performance "linking the individual, separately or simultaneously, with multiple audiences" (Papacharissi 2011, p. 304). Just as stage actors fulfill certain roles, people create and maintain different images for themselves which they 'perform' in accordance with social circumstances. In the case of social media, those images are 'performed' in accordance with the different platforms.

To better grasp the performance of identity on social media, it is important to understand how and why performances can change across each social media platform. Polymedia theory, as proposed by Madianou and Miller (2013), maintains that the proliferation and increased convergence of new media create an "environment of communicative opportunities that functions as an 'integrated structure' within which each individual medium is defined in relational terms in the context of all other media" (p. 170). Rather than only offering more communication options, polymedia enables the affordances of different media usages to be explored and negotiated by the users in terms of their emotional, social and contextual meanings (Willems 2021). Although this theory does not exclusively focus on social media platforms, we believe it is helpful to understand how different social media platforms are understood and used by young people in relation to their subjective identity and cultural identity performance. Our interest in (social media) platforms, moreover, is inspired by the strands of platform theory that explicitly connect the qualities of platforms to performance. van Dijck (2013a) notably argues that "a platform is a mediator rather than an intermediary, as it shapes the performance of social acts instead of merely facilitating them" (p. 29). The tools that platforms offer for this are called affordances and a range of studies have demonstrated that these differ significantly from one platform to another (Boczkowski et al. 2018; Boyd 2011; de Ridder and van Bauwel 2015). Meanwhile, media anthropologists have warned against a universal approach to social media affordances, arguing for a more context-sensitive optic through which to view the cultural specificity of social media practices (Costa 2018; Willems 2021; Miller 2016). Aiming to contribute to the increasingly intricate understanding of how such affordances of social media platforms relate to cultural identity, we will now turn to the Bolivian context.

### 2.2. The Bolivian Context

As pointed out by Rasmussen (2021), "Bolivia's history has spoken, and continues to speak, about how to understand and do difference." In scholarly literature on Bolivian cultural identity, the concept is generally presumed as a multiplex of affinities and belongings (Dangl 2019; García Linera 2014). Hence, Bolivianness is not only defined in terms of national belonging, but also through association with other cultural communities at different scales. Specifically, indigeneity, regionalism, colonialism, nation-making and globalization serve as important "cultural groundings", in Yin's (2018) terms, for cultural identity positioning in Bolivia.

For a considerable part of the Bolivian population indigenous identities, such as Aymara, Quechua and Uru, provide a source for self-understanding, self-expression and self-assertion. Recent census data indicate that 30% of the total population speaks an indigenous language such as Quechua, Aymara, and Guaraní (INE 2016) and more than 40% identifies as part of an indigenous group (INE 2015). Hence, indigenous identities in Bolivia are still very prominent, not as ossified articulations of the country's pre-colonial past and culture, but as contemporary ways of living, both in the remote regions of the country and

the cities (Arocena 2008). Especially, the recent influx of indigenous people in the cities re-configures the original meaning of indigeneity as something that is exclusively tied to rural territorialities (Albro 2010), although this is still a common perception both within and outside Bolivia (Rasmussen 2021; Haynes 2019). To position oneself as indigenous is accordingly not only a matter of language use (Canessa 2007) or apparel (cf. *pollera*, the typical colorful skirts; *llu'chu*, the brightly colored stocking caps) (Fabricant 2012), but also of growing political awareness about the marginalization of indigenous people as well as an active recognition and recovery of historical continuity, collective memory and indigenous ancestry (Albro 2005).

Partly related to indigeneity, regionalism also serves the role of a 'template' against which Bolivian people articulate their cultural identity. In a country with enormous regional variation in terms of natural geography, political history (Spanish colonization, indigenous movements), urbanity and rurality and related economic prosperity, regions such as the highlands (Altiplano) and tropical lowlands (Amazonia) are used as identity markers. In particular the use of terms such as *collas*, referring to people coming from Altiplano and cities such as La Paz, Oruro and Potosí, and *cambas*, used to designate people from the lowlands and cities such as Santa Cruz, Beni, and Pando, can be seen as symbolic vehicles for constructing socio-cultural distinctions between these regions and their inhabitants (Albó 2007).

The political aspects of Bolivia's history of nation-making have been widely discussed in the literature. The integrationist mestizo nationalism that was adopted by Latin American countries in the beginning of the 20th century—known as the strategy of mestizaje or the mixture of indigenous with non-indigenous people—also took root in Bolivia (Rasmussen 2021). Although mestizaje was initially adopted as the political strategy to create a sense of national unity, the ideology of multiculturalism underpins, since the 1990s, the construction of a Bolivian national identity, based on the idea of unity in difference (Sanjinés 2004). The MAS government and former president Evo Morales particularly have played a pivotal role in this quest to harmonize multiculturalism with a shared definition of Bolivian nationhood (Arocena 2008; Canessa 2007). Although Morales' role in pluri-national state reform has received much support among Bolivian people, his escape from the country in 2019 has exposed the tensions and intersections between class and race ((Rasmussen 2021; Albro 2010, 2005).

More recently, with global media and consumer culture, people perform their cultural identity in relation to the social imaginary of what it means to be a "citizen of the world" (Miller 2002). In Global South countries, this is especially found among students, educated or socially privileged people who want to come closer to the modern, sophisticated and affluent life they aspire to (Uimonen 2013). In Bolivia, people belonging to the social elite also identify more easily with a cosmopolitan than an indigenous way of living (Kollnig 2020), with the aim of participating in global and Western culture (Molz 2016). It has been found that Bolivian university students position themselves as global citizens in attaching importance to learning foreign languages, making friends from abroad, keeping themselves informed about international news and travelling abroad (Carrasco Alurralde and Oberliesen 2013).

## 3. Method

### 3.1. The Current Study

The purpose was to examine how Bolivian university students engage with social media in performing cultural identity. We conducted focus groups with Bolivian young people to qualitatively explore their cultural identity performances on social media. Using polymedia as the theoretical optic for understanding social media use (Madianou 2020; Madianou and Miller 2013), we were interested in how students engage with the performative affordances of all the social media that play a role in their lives, and their culturally-specific interpretations of the technological propensities of these social media.

Hence, rather than focusing on discrete platforms, the polymedia approach "shifts our attention to how users treat media as integrated environments" (Madianou 2020).

### 3.2. Participants

At the time of the study, participants were Bolivian young people aged 18 to 25, attending a Bolivian university. Given that we aimed for a diverse group of participants, we chose to recruit students from universities located in different regions. The Bolivian university landscape consists of state-subsidized public universities, attracting more than half a million students, and private universities with nearly 130,000 students enrolled (INE 2018). There have been references to the differences between public and private universities in terms of students' socio-economic milieu and political engagement (Carrasco Alurralde and Oberliesen 2013; Hastie Falkiner 2015). Students from private universities often grow up in more privileged middle-class and high-income families that are able to afford an expensive education, whereas students from public universities mostly come from medium- and lower-income families (Carrasco Alurralde and Oberliesen 2013). To fully comprehend the stratified effects of online communication and how much it might exacerbate gaps, polymedia activities in contexts of inequality and scarcity must be studied, as Madianou (2021) explains. Considering the differences between the university types and the potential role of a metropolitan context on cultural identity formation, we concentrated on the three biggest cities of Bolivia: Cochabamba, La Paz and Santa Cruz de la Sierra. Although these three cities are known as the so-called central axis, they are located in three different regions: central Bolivia (Valley or Valle), west-central Bolivia (Andean or Altiplano) and eastern Bolivia (Tropical or Oriente). Gender balance was a final important selection criterion.

All participants were recruited by the first author, a Bolivian PhD student based in Brussels, Belgium. Some were mobilized through social media postings on personal Facebook and Instagram accounts and via a flyer on Facebook groups where Bolivian students post information about their university and education. Most of the participants, however, were recruited in person on their respective campuses, where the first author distributed flyers and explained the purpose of the study. Once students agreed to participate, they were added to a WhatsApp group to keep them posted about any last-minute changes. The potential of using smartphones in qualitative research, especially when working with young people, was fully explored. It allowed the researcher to interact in a more informal way with the participants, when presenting herself and the research project (Kaufmann and Peil 2020).

### 3.3. Data Collection

We organized seven focus groups with public university students and 10 focus groups with private university students. Six focus groups were conducted in La Paz, five in Cochabamba and six in Santa Cruz de la Sierra. In total, 10 universities spread over 13 campuses were involved (see Table 1). Altogether, 60 young women and 45 young men participated.

All focus groups were conducted by the first author between December 2018 and March 2019 in Spanish, the most-spoken official language in Bolivia (INE 2016) and the language used in university education. Focus group interviews are often used in research on cultural identity, because they allow for group discussion through which identities are performed, produced, negotiated, affirmed and challenged (Clair et al. 2018). Given that the researcher grew up and studied in Bolivia and was not that much older than the participants, the focus group interviews were conducted in the spirit of constructive dialogue and symmetrical communication.

The main aim of the focus group was to learn from the students' perspectives how they defined their cultural identity and how they engaged with social media as a stage for cultural identity performance. The duration of the interviews ranged from 72 to 121 min, with a mean of 111 min. The interviews followed a script with four discussion topics in order to learn how they engaged with social media: (1) what meaning they gave to

cultural identity and Bolivian cultural identity specifically; (2) how they 'played' language differently offline and online; (3) how they performed their Bolivianness on social media.

**Table 1.** Overview of the research sites.

| City | University | University Type | Title 4 |
|---|---|---|---|
| Cochabamba | Universidad Católica Boliviana San Pablo | Private | FG5 |
| | Universidad Privada Boliviana | Private | FG6 |
| | Universidad del Valle | Private | FG7 |
| | Universidad Mayor de San Simón | Public | FG8, FG9 |
| La Paz | Universidad Mayor de San Andrés | Public | FG1, FG2 |
| | Universidad Católica Boliviana San Pablo | Private | FG3, FG4 |
| | Universidad Pública de El Alto | Public | FG10 |
| | Universidad Privada Boliviana | Private | FG11 |
| Santa Cruz de la Sierra | Universidad Católica Boliviana San Pablo | Private | FG12 |
| | Universidad Tecnológica Privada de Santa Cruz | Private | FG13 |
| | Universidad Autónoma Gabriel René Moreno | Public | FG14, FG15 |
| | Universidad de Aquino Bolivia | Private | FG16 |
| | Universidad Privada de Santa Cruz de la Sierra | Private | FG17 |

*3.4. Data Analysis*

All focus groups sessions were audio and video recorded, fully transcribed and annotated based on in situ observations and video footage. The interview transcripts were systematically coded with qualitative data analysis software in following a framework that contained two axes. One axis used the different social media as a point of entry. This axis functioned as the polymedia optic for identifying when how, and where the various social media experiences mobilized ideas about cultural identity performance. The other axis aimed at which forms of cultural belonging, affinity and identification the students referred to. This axis functioned as the cultural performance optic for analyzing how and when the participants invoked examples of cultural identity.

All interviews were examined to ascertain patterns of cultural identity performances on social media. Given the amount of data (159,371 words of transcripts), we used data analysis software to obtain an integrated overview of how social media experiences were combined with ideas about cultural identity. Inspired by internet-related ethnography research (Postill and Pink 2012), specific examples of social media posts that were discussed by the students were sometimes brought in to better understand the nuances of their identity performances.

**4. Findings**

*4.1. General Social Media Use*

Before going further into the qualitative analysis of cultural identity performances and social media, let us briefly consider how Bolivian students navigate the social media landscape for everyday communication and sociality. The available formal statistics on social media use in Bolivia indicate that young people are the most avid internet users. Most young people go online via their mobile phone. The most popular social media among Bolivian young people are Facebook, WhatsApp, YouTube, Twitter and Instagram (AGETIC and UNFPA 2019).

Students in our sample considered WhatsApp as the most popular and multifaceted hangout space to socialize with friends, acquaintances and to share information for academic purposes. Whereas virtually all students had a WhatsApp account (100 out of 105), Facebook was mentioned by 82 students as a place where they liked to meet up with classmates, friends and relatives. It is important to consider that we have not included Facebook Messenger in the focus groups discussions because the participants focused more on the public activities shown on Facebook as a social media platform. Unlike young people in

western societies, Instagram and YouTube appeared far less embedded in their everyday social media practices (respectively 61 and 44 students). Instagram was mainly discussed as a platform where one would follow "famous people", such as vloggers and influencers, or to connect with communities of interest, such as fashion, fitness, music, or travel. None of the students uploaded videos on YouTube, but used this platform to watch videos. Specifically, Latin American vloggers, such as the Mexican YouTuber Luisito Comunica and Ecuadorian comedians from Enchufe TV, were named as a reason to "hang out" on YouTube. Other social media such as Snapchat, Twitter, Pinterest, Reddit, TikTok, Line and Google+ were mentioned by less than eight participants and were thus scarcely touched upon in the focus groups. Especially, the relative insignificance of Twitter struck us the most. In light of the literature on the role of Twitter in Latin American protest movements, and Bolivian students' engagement with the political turbulence in their country (Hastie Falkiner 2015), we had expected Twitter to be discussed more. Through the polymedia lens, the autonomy of the participants in managing relationships according to the platform and the meanings of each platform in terms of usage were evidenced (Madianou 2014; Boczkowski et al. 2018).

### 4.2. Self-Evidence, Everydayness and Normativity

Another unexpected result was the students' initial tendency to be quiet regarding their cultural identity performances on social media. In fact, we found that explaining and illustrating what it means to perform cultural identity on social media was not at all evident for the students. Given that questions of culture, indigeneity and nationhood were central in the political context in which these young people grew up, we had supposed that the focus group discussions would easily take off. However, this was not the case. There are several possible explanations for this. A first might be that cultural identity performances are so "grounded in the everyday, in the mundane details of social interaction, habits, routines, and practical knowledge" (Edensor 2002), that they pass by without being noticed. Specifically in a country such as Bolivia where the expression of cultural identity, in its plural meaning, determines the streetscape, political and public debate, this might be the case.

A second could be that with mobile communication devices a large part of our engagement with social media has become so ubiquitous that it has become "an ambient background of indirect communication" (Madianou 2020). Hence, the students were asked to reflect on everyday experiences that remain in fact largely invisible, unless they find themselves in situations where both their cultural identity and media use are de-familiarized. That is the effect the focus groups probably had on the students.

A third possible explanation can be found in how the students wanted to project an image of themselves. Looking at the discussions as meaningful sites for cultural identity performances in their own right, many students initially told us that they were indeed engaged with cultural identity but that they did not give it priority in their online posting and sharing practices. We believe this was part of the general tendency among Bolivian university students to position themselves as global citizens (cf. supra), of which sophisticated social media use is perceived as an integral part (Sinanan 2020). The interaction with the researcher who conducted the focus groups, a fellow Bolivian citizen who was doing her PhD in a European country, might have reinforced the students' tendency to articulate a sense of global citizenship and consumerism, emphasizing that they were not so different from 'ideal-type' social media users. In the post-it sessions and via the use of visual elicitation materials, however, we were able to go beyond the self-evidence, everydayness and normativity that initially set the tone.

### 4.3. Cosmopolitan Identity Performances

The largest part of the examples that students found worthwhile to elaborate on were related to how they used social media to stage their cosmopolitan way of being, their know-how about global mass culture that remains centered in the West and their engagement with this. All five social media that were discussed in the focus groups were

mentioned as platforms where they presented themselves as oriented to the wide world, performing "a refined cosmopolitan" (cf. Dumitrica 2019). In fact, this occurs because social media facilitates the development of cosmopolitanism (McEwan and Sobre-Denton 2011). Using English on Facebook and Instagram, the social media that were felt the most public-facing (cf. Miller 2016; Costa 2018), proved to be considered as a self-evident choice to tailor their performances for more general international publics, outside Bolivia and the Latin American continent. Students expressed their appreciation for foreign culture and provided many examples of how they performed their 'world-savvy' taste in food, music, art and clothing on social media. Facebook, especially, was mentioned as a signboard for their "global style youth life" (cf. Rygaard 2003). They said that they watched cooking videos on making sushi and Italian pasta. Fandom proved to be an important cosmopolitan identity marker. The ubiquitous K-pop phenomenon also popped up as an example of their cosmopolitan consumer orientation in music and clothing, especially among female students from public universities. The other examples of musical artists they engaged with on social media came from the United States and Europe. Among male students, supporting soccer teams (such as Real Madrid FC, Barcelona FC and Milano) through posting videos and pictures were presented as signs of worldliness.

Our findings dovetail with other studies on the use of social media as a place where people who do not live in the Global North can curate and aspire to a cultural identity that is in line with the ideal of an international, cosmopolitan and fashionable self (Miller 2016; Jakaza 2022). For example, Maria (FG14), explained how she displayed her passion for Italian culture and New York city on social media: "Italian art, everything that is Italy in reference to painting, or food. I love Italian food! From New York, I share content of the city's streets, fifth avenue, the big apple . . . "

Social media posts about iconic cities, such as New York and Paris, well illustrate how social media are not only used to express aspirations but also to reveal the unattainable dream of international travelling for many young people in Bolivia. The meme below (see Figure 1) is a good illustration of this. Maria shared it during our research to display her desire to visit New York. In a Latin American context and Bolivian context specifically, it is no coincidence that the person on the right has darker skin than the actress on the left. It implies significant differences in socio-economic status between the happy few for whom international travelling is in reach, and many others who can only dream of this. Our material confirms Miller's (2016) observation that humor and irony are often used to puncture the pretensions of the cosmopolitan elite, while articulating how international experiences are craved.

International travelling, seen as a token of cosmopolitan orientation, is out of reach for the majority of Bolivian students. However, by participating in the massively available visual material on iconic world cities such as Paris, London and New York, they were projecting a cosmopolitan sensibility. Facebook proved to be a place that encouraged this kind of socially acceptable performance, whereas YouTube allowed them to follow tourist influencers. Some students who had actually travelled abroad used Instagram to evidence their trip and found it important to present themselves through stylized visual posts of landscapes and tourist attractions.

A further example that students often mentioned when discussing the culture–social media nexus were festivities such as Saint Valentine's Day, Halloween and Thanksgiving. Borrowed from the United States and, not coincidentally, three festivities that are increasingly commercialized and mass-marketed, all three were pointed out as special calendar days that they would celebrate on social media. Bolivia has many living cultural traditions and festivities that go back to pre-colonial indigenous cultures as well as the Spanish colonial system, but US traditions were generally accepted as international traditions one has to engage with in social media. Hence, although All Saints' Day is still widely celebrated in Bolivia, students indicated that on social media Halloween was more fervently embraced as the popular festivity of death and life.

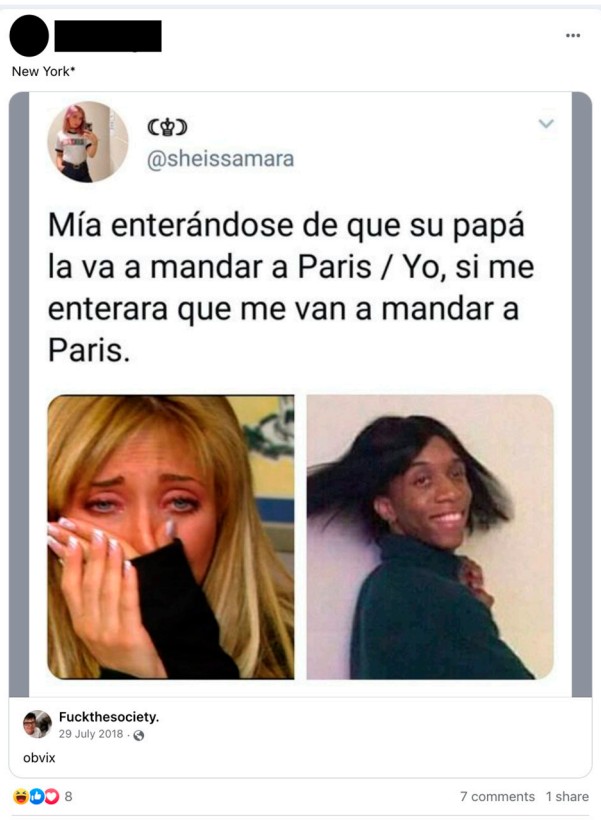

**Figure 1.** If I were sent to New York. (meme). The meme features, on the left, Mia, a character from a popular Mexican telenovela, in tears, whereas on the right we can see a joyful, happy person. The accompanying text states: "Mia finding out that her father is going to send her to Paris/Me, if I would find out that I would be sent to Paris". Maria, the participant, added "New York" in her post because she would like to go to New York instead of Paris. This is an example of how her reaction would be if she would be Mia.

### 4.4. An Antagonistic Articulation of Bolivianness

Students choose particular social media and social media genres to articulate their allegiance to Bolivia as their country. In tune with polymedia theory, we found that the perceived scalability and communicative affordances of the different social media used in their everyday life, such as the size of the public they are engaging with, the degree of intimacy they can curate and the type of social and emotional communication they want to achieve (Miller 2016; Madianou and Miller 2013), encouraged antagonistic expressions of what it means to be Bolivian. Hence, the enactment of national sentiments oscillated between self-glorification, pride and boosterism, on the one hand, and critical, ironic, and sometimes even self-denigrating performances of Bolivianness on the other.

Whereas only a few references to global politics were made, national politics on social media triggered quite animated discussion. In this respect, memes were one of the most resonating genres that were mentioned. Even though the focus groups took place almost three years after the hotly contested referendum of 21 February 2016, in which 51% of Bolivians voted against the fourth candidature of Morales for the presidential elections, many students referred to examples of memes they had shared on Facebook and Twitter at that time. Posting, sharing and commenting on Bolivia's political crisis emerged as significant practices through which they acted out their concern about Bolivia and the way their country's political situation was perceived in neighboring countries and further afield.

As a reaction to the image of their country as politically unsettled and culturally backward, for instance in memes from neighboring countries such as Chile (Haynes 2019), some students emphasized that they felt it important to display Bolivian achievements on

their social media accounts, for example, by posting news about successful Bolivian athletes and scientists as sources of national pride. Other topics, such as Bolivian gastronomy and culinary tradition, that have a high degree of identification and commonality, were referred to as examples of how they would articulate their Bolivianness and national self-esteem, as Daniela, a participant from FG1, explained.

> Perhaps (on social media) we highlight activities of Bolivian people that are worth being shared, such as winning a medal, representing (Bolivia) in an international competition and say "that's something Bolivian to be proud of". (FG1)

Students explained that they also envisaged the wider global audience by posting, linking and sharing pictures and videos of typical tourist places in Bolivia, such as the Uyuni Salt-flats, Lake Titicaca and Copacabana, on Instagram and Facebook in particular. In this matter, students also referred to Facebook video posts on traditional festivities and ceremonies, such as the Carnival of Oruro and the Day of the Dead. Some students stressed the importance of English in bringing the beauty of their country and cultural traditions to the attention of international audiences.

Next to these examples that can be categorized as promotional or self-glorifying national identity performances, students provided numerous examples of more playful performances of Bolivianness on social media. Memes proved to be a genre that was used to mock serious performances of national belonging, as we discussed earlier. Topics varying from Bolivian traditions, customs and cultural events to controversial political issues, such as the referendum of 2016, were fuel for memes that were shared on Facebook accounts and WhatsApp stories. It was clear that the audiences that they had in mind oscillated between the intimate (on WhatsApp) and the semi-public (on Facebook). WhatsApp in particular emerged as the sheltered place where this kind of self-reflexive, ironic performances were shared, enjoyed and served as pivotal tools for bonding with peers. For example, making fun of their Bolivian-accented Spanish as opposed to other Latino idioms was a practice the students often referred to. Other examples of common banter we heard in the focus groups related to typical traditional Bolivian dishes that only Bolivian people would know and eat. Hence, the act of sharing the meme hereunder (see Figure 2) was described as a way of playfully engaging with food traditions that distinguish Bolivia from other Latin American countries.

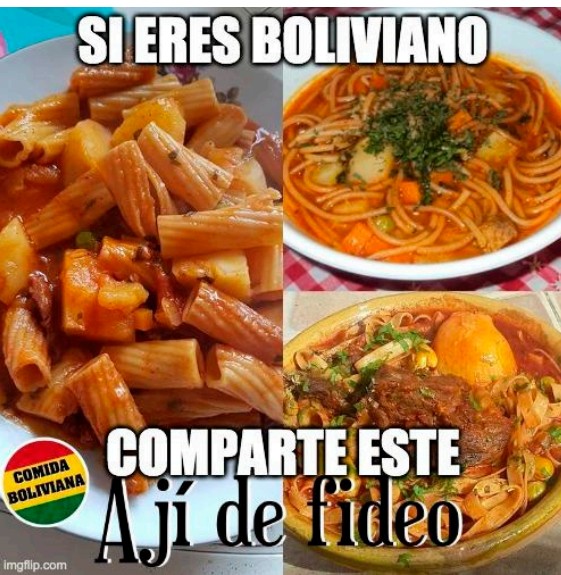

**Figure 2.** Share this *ají de fideo* (meme).

*4.5. Indigenous Identity Performances for Intimate Audiences*

Whereas hardly any student explicitly identified with one of the indigenous cultures, as also observed by Haynes (2016), affinity with regions and departments was more overtly

expressed in the focus groups. Cultural allegiance at a regional and departmental scale can hint at feelings of indigenous belonging, since specific indigenous groups historically have their roots in distinct territorialities. Some of the students, mainly from public universities, stated that they liked to engage with content related to the Aymara and Quechua cultures, recognizing they had indigenous roots without formally identifying with these cultures. The importance of sharing indigenous traditions on social media popped up in the example of k'oa, an originally Andean ritual that today, in Bolivian cities also, is celebrated during Carnival season within the family circle. Another way of performing indigenous identity was shown in the subtle use of indigenous language. For example, students talked about images they posted containing some words in Quechua with a translation next to it. Again, numerous examples they gave were memes, rendering these cultural identity performances playful and mock serious, as the following statement from Melba, one of the participants in FG10, illustrated:

> I always share those memes about yatiris (i.e., indigenous priests who read coca leaves as a divination practice) saying: "It's evident my dear, you are going to drink alcohol the whole weekend, and Monday as well" (see Figure 3). Because they are so right . . .

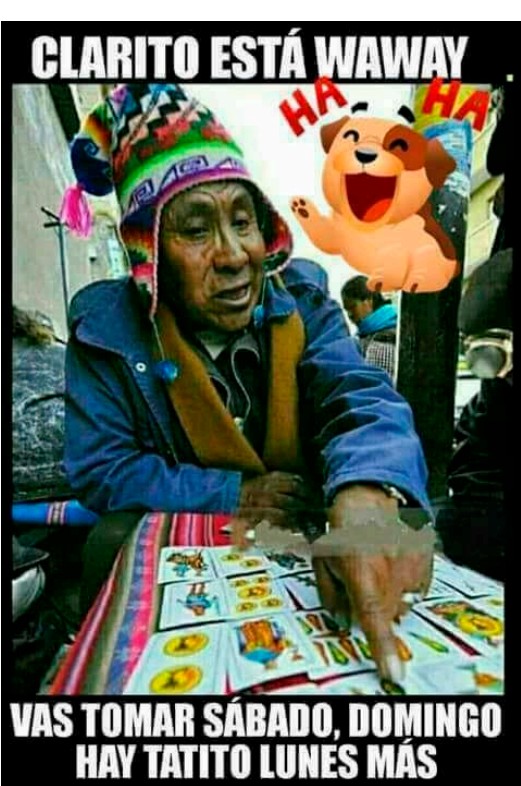

**Figure 3.** It's evident my dear (meme).

Students gave many other examples of Facebook memes in which the differences between people coming from different regions in Bolivia were illustrated. These memes highlighted cultural disparities and idiosyncrasies regarding culinary traditions, cultural customs, idiom, personality traits and geography. Hence, these memes can be understood as vehicles to 'do difference' in a country that has been searching for new definitions of national unity in recent decades. Students often indicated that they retrieved these memes first from Facebook and then shared them on their WhatsApp stories where only their culturally proximate contacts could see them.

It was only when indigenous identity performances were discussed that individual negative experiences came to the surface. A few students, such as Milka in FG1, said that

they had posted their status on Facebook in an indigenous language, but that this was not regarded as appropriate by their Facebook friends.

> I post songs in Aymara because . . . We learned the song Rice Pudding in Aymara, which I found to be a very tender song. So, I used this and published it on Facebook, since my name is Milka and the title of the song is arusamp millk'impi (rice pudding). But then they (my friends) told me: "Ah, are you mentioning yourself in the song?" And they started to bully me, just for that. But I posted it because it seemed tender to me.

It is not certain if an experience such as Milka's is an indication of ethnic discrimination and social exclusion, but it is remarkable that students who demonstrated adherence to Bolivia's indigenous heritage (e.g., stories, songs, folklore, cultural values, rituals, language, etc.), did not feel encouraged on social media to express this. It is perhaps not by accident that WhatsApp, as the most intimate platform among their social media suite, was considered as the only place where they could possibly calibrate forms of indigenous belonging more carefully and authentically without running the risk of being culturally misunderstood, declined or mocked as 'backward' or 'opportunist'.

## 5. Conclusions

With mobile communication, social media have become more ubiquitous in everyday life, choreographing and proposing the stages and the publics for whom persons can perform their cultural belongingness, especially among young people (Dumitrica 2019). The contingency for tremendous visibility and scalability, together with the possibility of a more controlled, as well as more aspirational, presentation of the self (Papacharissi 2011), begs the question as to how cultural identities, which revolve around sharing and allegiance, are recollected, recontextualized and refashioned in the very act of performing before "networked publics" that are both global and local (Uimonen 2013). This article departed from the observation that there is increased scholarly interest in the experiences of cultural identity on social media in Global South contexts. Rather than viewing technological affordances in essentialist terms, we have tried to understand the role of social media from the specific cultural context in which they are used. Key debates in this context have revolved around the global–local nexus in mass culture and the way in which social media potentially play a role in revitalizing and re-articulating cultural traditions and identities. As these debates are still in need of further empirical evidence from particular Global South settings, we have turned to the case of Bolivian university students and how they understand social media's role in representing their cultural affinities and belongings. With mainly formal statistics on social media use in Bolivia, our research provided exploratory qualitative data on the popularity of particular social media among young people in Bolivia and their everyday social media practices. Looking at their experiences from an everyday perspective, our study has taken inspiration from ethnographically-oriented social media research as well as polymedia theory (Madianou and Miller 2013). Furthermore, the recent political history of Bolivia, with its dynamics of regionalism, indigenous politics and nation-building intersecting with the social elite's aspiration for cosmopolitanism, has also led us to highlight literature on the specific Bolivian context.

The findings of this study can contribute to the ongoing efforts to understand the entanglements of cultural identity and social media from the perspective of young people beyond Global North contexts. The various belongings and attachments that are part of cultural identity for Bolivian students, ranging from regional and indigenous to national or cosmopolitan, are not actively associated with or performed on all social media, despite social media platforms being omnipresent in their daily lives. The perceived scales of visibility (from more public to more private), intimacy (from a space shared with close ones to one shared with strangers) and normativity (e.g., the tendency to emphasize cosmopolitanism as part of social aspirations) of these platforms shape their cultural identity performances.

We believe there are at least two specific contributions to the interlocking fields of polymedia and identity performance on social media. Firstly, besides the role that perceived visibility, intimacy and normativity play, the (perceived) specificities of genres sometimes seem to override platform affordances. Some social media genres, particularly memes, seemed particularly suitable for articulating views and experiences regarding politicized identities such as national identity for the participants. What they post about their Bolivian cultural identity reflects ambivalent feelings of what it means to be Bolivian. Particularly, the genre of memes is appropriated as a way of expressing Bolivianness with detachment. Through more public platforms such as Facebook, these memes can be drawn into more intimate ones such as WhatsApp, despite the platform specificities, hence showing the limits of traditional polymedia theory. Secondly, the study demonstrates that social media pose an important stage for expressing cultural identity, specifically with the purpose of distinguishing oneself from others. This is the case from a national point of view, whereby social media would be used to promote a positive image of Bolivia in times when it is suffering from a bad international reputation. For social elites, which university students tend (aspire) to belong to, it also seems to be more important to articulate publicly their cosmopolitan rather than indigenous belongings. Their polymedia attitudes seem to amplify cosmopolitan performances and at the same time they reduce local performances. Their indigenous belongings thus find their way to more private social media spaces and they were mainly expressed by the students from public universities. This study found that social media are important sites for Bolivian students to perform their cultural identity, and that they engage with this in various ways: from playful to serious, from promotional to political. The results suggest that, although indigenous 'ways of being' are part of their cultural identity definition, Bolivian students do not articulate this strongly in their social media usage, and certainly not on platforms offering public moments to a wide audience. In contrast, a strong attachment to cosmopolitan identity seems to be prioritized in the different cultural identity positions they are performing online. This is particularly important in the current political moment in Bolivia, i.e., a country struggling between unity and division, between sameness and difference. Social media thus seem to be crucial resources to highlight identities in a politically divided context. This would warrant further research in other contexts and from the vantage point of other social groups in order to better understand the specific workings of cultural identity performance on social media. The students involved in this research can be considered as belonging to a privileged group of young people who can pursue university studies. One possible explanation for their disposition towards cosmopolitanism and playful Bolivianness might be their aspiration to social and economic status. This aspect has been under-explored in this study, but definitely needs more attention. Especially the question as to how social media platforms such as Facebook, YouTube and Instagram are capitalizing on these feelings is worth investigating.

**Author Contributions:** Conceptualization, P.C.S., J.B. and K.S.; methodology, P.C.S.; software, P.C.S.; validation, P.C.S., J.B. and K.S.; formal analysis, P.C.S.; investigation, P.C.S.; resources, P.C.S., J.B. and K.S.; data curation, P.C.S.; writing—original draft preparation, P.C.S. and J.B.; writing—review and editing, P.C.S., J.B. and K.S.; visualization, P.C.S.; supervision, J.B. and K.S.; project administration, P.C.S.; funding acquisition, P.C.S. and J.B. All authors have read and agreed to the published version of the manuscript.

**Funding:** This research was funded by a PhD bursary financed by the development cooperation research program of the Vrije Universiteit Brussel (2017–2021).

**Institutional Review Board Statement:** Not applicable.

**Informed Consent Statement:** Informed consent was obtained from all subjects involved in the study.

**Data Availability Statement:** Not applicable.

**Conflicts of Interest:** The authors declare no conflict of interest.

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
