# Peer review of "Cultural Identity Performances on Social Media: A Study of Bolivian Students"

_journalmedia, doi:10.3390/journalmedia4010021_

Round 1
Reviewer 1 Report
Cultural identity performances on social media: a study of Bolivian students
This is a nice and well-written article about the performance of cultural identity on social media among university students in Bolivia. The article combines polymedia theory and cultural identity theory to create a framework for analysing focus groups conducted with students at public and private universities in Bolivia. Although the article is neatly structured and clear, I do have some suggestions for further improving the text:
The theoretical framework is somewhat superficial, and the two different theories used are neither discussed in relation to each other, nor particularly deepened or discussed by/in the study. Polymedia theory for example seems like a very suitable approach for analysing social media as an environment, but the theory is not really discussed or debated in the text, and hence just remains a way for concluding that young people use different social media platforms for different purposes.
The method for gathering and analysing the empirical material is adequate, but I do have a few remarks. Firstly, that the empirical material only consists of students, but that this is not really taken into account in the analysis. There is a huge difference between university students and other young people, and it is not meaningful to just treat them as any young people (as is done here). There is also a distinction made between students at public and private universities in the methods section, but this distinction is not particularly used in the analysis. Why has the author(s) selected students from two different kinds of universities if it is not important for the analysis? I also reacted upon that the author(s) both indicated that the respondents in this study represented a whole generation (they certainly do not, due to the qualitative character of the study, as well as for reasons discussed above), and that the results at some points were presented as per cents.
The analysis is descriptive, and the author(s) should try to deepen it and draw some theoretical conclusions from it.
Reviewer 2 Report
The paper is in good shape, overall. I would only add a clear definition of social media in the first part. For example not all scholars agree that WhatsApp or YouTube are social media platforms, so it would be good to hear how the author(s) see it. Moreover, when speaking about Facebook it is not clear whether the author(s) also take Messenger into account or treat is a different app/platform.
I am not sure if the author(s) have an official permission to share the participants' names, but I would anonymize Maria Claros' name that appears on one of the illustrations (page 7).
